Alleviation of drought stress by root-applied thiourea is related to elevated photosynthetic pigments, osmoprotectants, antioxidant enzymes, and tubers yield and suppressed oxidative stress in potatoes cultivars

Saleem Muhammad Hamzah 1
http://orcid.org/0000-0001-5638-809X Wang Xiukang 2 wangxiukang@yau.edu.cn
Parveen Abida 3 abidauaf@yahoo.com
Perveen Shagufta 3
Mehmood Saqib 3
Fiaz Sajid 4
Ali Sajjad 5
Hussain Sajjad 6
http://orcid.org/0000-0001-9081-2229 Adnan Muhammad 7
Iqbal Naeem 3
Alatawi Aishah 8
Ali Shafaqat 9 10
1 College of Plant Science and Technology, Huazhong Agricultural University , Wuhan , China
2 College of Life Sciences, Yan’an University , Yan’an, Shaanxi , China
3 Department of Botany, Government College University , Faisalabad , Pakistan
4 Department of Plant Breeding and Genetics, The University of Haripur , Haripur , Pakistan
5 Department of Botany, Bacha Khan University , Charsadda , Pakistan
6 College of Agronomy, Sichuan Agricultural University , Wenjiang, Chengdu , P.R. China
7 Department of Agriculture, University of Swabi , Swabi , Pakistan
8 Biology Department, Faculty of Science, Tabuk University , Tabuk , Saudi Arabia
9 Department of Environmental Sciences and Engineering, Government College University , Faisalabad , Pakistan
10 Department of Biological Sciences and Technology, China Medical University , Taichung , Taiwan
Mostofa Mohammad Golam
Electronic publication date: 2022 Apr 7
Publication date: 2022
Volume: 10
Electronic Location ID: e13121
Received 2021 Jul 5; Accepted 2022 Feb 24
Copyright: © 2022 Saleem et al.
Copyright year: 2022
Copyright holder: Saleem et al.
License: This is an open access article distributed under the terms of the Creative Commons Attribution License, which permits unrestricted use, distribution, reproduction and adaptation in any medium and for any purpose provided that it is properly attributed. For attribution, the original author(s), title, publication source (PeerJ) and either DOI or URL of the article must be cited.
License URL: https://creativecommons.org/licenses/by/4.0/

Keywords: Drought, Antioxidant, Yield, Potato, Osmoprotectants

Funding: National Key Research and Development Program of China 2017YFC0504704 National Natural Science Foundation of China 51669034, 41761068, 51809224 The publication of the present work is supported by the National Key Research and Development Program of China (grant no. 2017YFC0504704) and the National Natural Science Foundation of China (51669034, 41761068, 51809224). The funders had no role in study design, data collection and analysis, decision to publish, or preparation of the manuscript.

==============================
The growth and productivity of plants are enhanced by the use of thiourea (TU) under stressful conditions. When TU is applied as a rooting medium, it improves plant growth characteristics and other physiological parameters in stressed environment. A pot experiment was conducted in the botanical garden of the Government College University, Faisalabad 38000, Pakistan to examine the TU-mediated fluctuations in some crucial physio-biochemical parameters and the oxidative defense of potatoes under a restricted water supply. For this purpose, two potato cultivars (potato-SH-5 and potato-FD-73) were sown in pots containing 10 kg of soil. Water was regularly applied to the pots until germination. After 2 weeks of germination, drought stress with 65% field capacity was imposed, while the control was subjected to 100% field capacity. TU, as a rooting medium, was applied at the vegetative stage (0 (no application), 0.5, 0.75 mM). A substantial reduction in the total number of leaves, leaf area, tuber biomass (fresh and dry weight), photosynthetic pigments, membrane permeability, and leaf relative water content (RWC) was recorded in plants under drought stress conditions as compared to control plants. The damaging eﬀects of water stress were more critical for cv. potato-FD-73 as compared to cv. potato-SH-5. In contrast, drought stress enhanced the malondialdehyde (MDA) and hydrogen peroxide (H2O2) content while also increased antioxidant enzyme activities (superoxide dismutase (SOD), peroxidase (POD), and catalase (CAT)) and triggered the accumulation of soluble proteins, soluble sugars, proline, and phenolic and anthocyanin contents. However, TU applied as rooting medium at 0.5 and 0.75 mM was eﬀective in reducing the detrimental eﬀects of water stress in both cultivars. Furthermore, increasing levels of TU enhanced chlorophyll pigments, dissolved proteins, complete dissolved sugars, and enzymatic capabilities of POD, SOD, and CAT, while reducing the MDA and H2O2 in both cultivars under stress conditions. In conclusion, TU improved the yield and chlorophyll pigments of potato plants by mitigating the adverse effects of drought stress through reduced EL, MDA, and H2O2 contents and improved activities of enzymatic and non-enzymatic antioxidants and osmoprotectants.

Introduction

Plants are typically exposed to a broad myriad of biotic and abiotic stresses, including feeding from wild animals and insects, weed infestation, hail, mechanical injury, diseases, low soil fertility, drought, salinity and others that can diminish the plant photosynthetic area, and thus, the attained total plant biomass or grain yield (Ali et al., 2020b; Naz & Perveen, 2021; Parveen et al., 2021, 2019; Perveen et al., 2019). In the case of plants grown in a hot, arid or semi-arid climate two of the main yield and biomass-limiting stresses are salinity and drought (Alam et al., 2021; Ali et al., 2020a; Kamran et al., 2019; Yaseen et al., 2020). About 33% of the world’s agricultural land is facing water imbalance and promoting drought vulnerability, which may drastically decrease the growth and yield of cereal crops (Hussain et al., 2018; Nawaz et al., 2021). Drought stress drastically affects the development of leaves, enzyme activity, and ion balance and eventually causes a reduction in crop yield (Hameed et al., 2021; Mumtaz et al., 2021; Yasmin et al., 2021). Similarly, it severely affects the process of photosynthesis at crowns, respiration, stomatal conductance, and physiological functions by causing feedback inhibition of the transportation of chlorophyll and other photosynthetic products to sink organs (Javed et al., 2020; Parveen et al., 2020; Saleem et al., 2020a; Zaheer et al., 2020). Plants utilize antioxidants to mitigate the toxic properties of reactive oxygen species (ROS); therefore, antioxidants are used to scavenge ROS. However, the photo reduction of oxygen and production of ROS, together with the superoxide anion, hydrogen peroxide (H2O2), and the hydroxyl radicals (Afzal et al., 2020; Hashem et al., 2020; Heile et al., 2021; Imran et al., 2019; Kamran et al., 2020; Rehman et al., 2019; Saleem et al., 2020d). Furthermore, some of the higher plants initiate the accumulation of osmolytes, such as glycine betaine, soluble sugars, proline, and many more secondary metabolites, to protect against drought stress (Ahlem, Lobna & Mohamed, 2021; Aziz et al., 2021; Hussain et al., 2016; Kaya et al., 2020; Naz & Perveen, 2021). From preventing membrane degeneration, enzymes and macromolecule’s lysis activation of the plant’s stress defensive mechanism is highly important for the plant’s survival.

Thiourea (TU) is an organosulfur compound with the chemical formula SC(NH2)2 and is structurally similar to urea, except that the oxygen atoms are replaced by a sulfur atom; nevertheless, the properties of both these compounds are different (Kaya, Ashraf & Sönmez, 2015). At physiological and organic levels, it rallies gas exchange, nutrient acquisition by root accommodation, and the recovery of the metabolism of sugar, subsequently increasing protein biosynthesis (Arafa & Ibrahim, 2018). These derivatives play key role in therapies viz., antibacterial (Venkatesh & Pandeya, 2009), anti-HIV (Venkatachalam, Mao & Uckun, 2004), anti-inflammatory (Keche et al., 2012), ligand and organocatalytic (Arafa & Ibrahim, 2018; Kaya et al., 2015; Naz & Perveen, 2021) anticancer, antioxidant, antitubercular (Zhao et al., 2013), and antifungal (Wang et al., 2019). In addition, these compunds and their derivatives are also used in herbela or agro-products and also used in entonomogy for better growth and devlopment in eco-friendly insects. Furthermore, TU has many beneficial effects such as it can increase the growth and yields of many plant species even when grown in stressed environment. When TU was applied as seed pre-treatment, it enhanced seed sprouting, and when it was used as a foliar spray, it upgraded the gas exchange as proved by many previous studies (Baqer, Al-Kaaby & Adul-Qadir, 2020; Khan et al., 2019; Yadav et al., 2020).

Potato (Solanum tuberosum L) is a herbaceous plant belonging to the Solanaceae family that grows up to 100 cm (40 inches) in length (Alkharpotly, Roshdy & Mady, 2018; Shirur, Channakeshava & Bhairappanavar, 2021). Potatoes produce tubers with a thickened stem and found in the Punjab province in Pakistan, especially in the area of: Okara, Sahiwal, Gujranwala, and Sheikhupura (Shirur, Channakeshava & Bhairappanavar, 2021). Potatoes have become the fourth most significant crop in terms of yield and production volume (You et al., 2019). Many studies have been conducted to indicate that potato tubers comprise some phenolic compounds, such as chlorogenic acid, which have been shown to possess free radical-scavenging activity in vitro (Germchi, Behroozi & Badri, 2011; Lenka & Das, 2019; Zheng et al., 2018). A nutritional study of the potato revealed that it contains 79% water, 2% protein, and 17% carbohydrates (88% starch) and is a rich source of vitamin B6 (about 23%) and vitamin C (24%), while providing 322 kilojoules (77 kilocalories) of food energy (El-Dissoky & Abdel-Kadar, 2013). It is likely that the potato will become a leading crop and could be used to meet nutritional needs globally (Zheng et al., 2018). However, many areas of potato production in developing nations are located in semi-arid regions, where drought stress negatively affects the harvest, threatening the sustainability of the potato industry (Alkharpotly, Roshdy & Mady, 2018). Hence, there is a need to enhance potato cultivation in areas with periodic drought stress due to decreasing water resources. This research was carried out to explore whether the application of root-applied TU has the potential to sustain growth and yield of the potato by regulating morpho-physiological and biochemical traits under drought stress.

Materials and Methods

Plant material and treatments

The experiment was conducted during November in the Department of Botany, Government College University of Faisalabad 38000, Pakistan under the natural climatic conditions (average temperatures of day and night were 25 °C and 13 °C, respectively). Relative humidity (RH) ranged from 50% to 85%, and length of the day from 11–12 h. Tubers of two potato cultivars SH-5 and FD.73-110 were collected from Ayyub Agricultural Research Institute (AARI) Faisalabad, Pakistan. Five potato tubers were sown in plastic pots (25 centimeter height and 20 centimeters internal diameter) containing 10 kg silt loam soil with a pH ranging between 5.5 to over 7.5 and with electrical conductivity (EC) of 1.60 dSm−1. After sprouting of potato seedlings, pots were irrigated with canal water. The seedlings were irrigated with canal water served as control. After those 15 days old seedlings were exposed to two separate moisture regimes (65% field capacity as drought stress and 100% field capacity as control). At soil dry weight basis soil moisture was calculated. After 2 days’ interval to recompense the water deficiency by evapotranspiration the pots were weighed. Thus, the soil moisture of pots was kept at 65% and 100% field capacity regarding to the treatments. After 2 weeks of applying drought stress at 65% field capacity except control all seedlings were treated with rooting media of thiourea application (0.5 & 0.75 mM) as 25 ml solution from each concentration at vegetative stage. The potato samples were dried in oven at 75 °C for 2 weeks. Potato roots and leaves sampling was done after 15 and 30 days of application of the treatment. Data was recorded for growth, physiological and biochemical attributes at vegetative stage. The experiment was laid out in a completely randomized design with three replications. This study did not include any information regarding human sciences.

Growth attributes

Number of leaves were recorded by counting the numbers with general observation. Although, the leaf area was measured by using the formula of Carleton & Foote (1965). Leaf area was calculated by measuring the maximum width and length. Total leaf area = maximum leaf length × maximum leaf width × correction factor. Where correction factor v (C.F) = 0.72. The tuber fresh weight was measured using electrical weighing balance and expressed in grams. Later, tuber samples were dried in an oven at 105 °C for 1 h, then at 70 °C for 72 h to determine their dry weight.

Photosynthetic pigments

Photosynthetic pigments was calculated by the standard method of Arnon (1949).

Relative water content: After measuring the fresh weight of potato leaves which collected from each replicate. After that sample was immediately hydrated to full turgidity for 3–4 h at room temperature. Then dried the leave sample with tissue paper and obtain fully turgid weight (TW). After that sample was placed in oven at 80 °C for 24 h and weighted to calculated the dry weigh (DW). All weighing is done to the nearest mg (Flower & Ludlow, 1986).

Proline contents

Proline content: The free proline content was determined on the basis of standard curve at 520 nm absorbance and expressed as µmol (g FW)−1 (Bates, Waldren & Teare, 1973).

Oxidative stress indicators

Malondialdehyde (MDA): Lipid peroxidation in the cellular organelles of potatoes cultivars was calculated using the determination of malondialdehyde (MDA) contents in the leaves. Lipid peroxidation was measured according to the method described by Heath & Packer (1968).

Hydrogen peroxide: For the analysis of hydrogen peroxide (H2O2) in the leaves of the potatoes cultivars, an H2O2 Assay Kit (Suzhou Comin Biotechnology Co., Ltd., Suzhou, China) was used as presented by Jana & Choudhuri (1981).

Osmolytes

The residue was prepared for extraction of osmolytes with 50 mg of dry weight of the plant dissolved in 10 ml of ethanol (80%) and the determination of total soluble protien (Bradford, 1976), phenolics (Bray & Thorpe, 1954), anthocyanin (Lewis et al., 1998) and total soluble sugars (Dubois et al., 1956) was performed from the extracts.

Antioxidant enzymes

Superoxidase dismuatse: The activity of superoxidase dismutase (SOD) was measured according to the method of Chen & Pan (1996) and expressed as U g−1 FW. Peroxidase (POD) activity from leaves was measured following the protocol of Sakharov & Ardila (1999) using guaiacol as the substrate. The activity of catalase (CAT) was measured according to the method of Aebi (1984).

Statistical analysis

The present findings were analyzed using analysis of variance (ANOVA) by using COSTAT software (Koch et al., 1982), and least significance difference was determined at a 5% probability level. Graphical presentation was carried out using Origin-2017 software. RStudio was used to illustrate the correlation and principal component analysis.

Results

Changes in the total number of leaves and leaf area

Drought stressed environment was significantly reduced the total number of leaves per plant and leaf area in both cultivars of potatoes (potato-SH-5 and potato-FD-73), compared to the plants grown in the control treatment (Fig. 1, Table 1). Although, the negative impact of drought stress can ameliorate by the foliar application of TU in the controlled plants and also drought stressed plants (Fig. 1, Table 1). Although, increasing levels of TU significantly increased plant growth and biomass in both cultivars of potatoes. From the present results we have noticed that the two potato cultivars, cv. potato-SH-5 and potato-FD, showed an increased in the total number of leaves by 20.32% and 24.25% and leaf area by 17.20% and 29.12%, respectively, compared to the cultivars which were grown in the controlled treatment. Although, the number of leaves were increased by 5.75% in potato-SH-5 and 6.33% in potato-FD-73 when applied the foliar spray with TU, compared to the plants which were not applied by the TU. The overall growth and biomass parameters are showing that the cultivar cv. potato-SH-5 showed better growth compared with the potato-FD-73.

Figure 1 Effect of root applied TU on (A) number of leaves, (B) tuber fresh weight, (C) leaf area and (D) tuber dry weight of two potato cultivars (Potato-SH-5 and Potato-FD.73-10) grown in controlled (100% water field capacity) and drought (65% water field capacity) stressed environment.

Means sharing similar letter(s) within a column for each parameter do not differ significantly at p < 0.05. Data in the figures are means of three repeats (n = 3) of just one harvest of potato plants ± standard deviation (SD). Different treatments (TU) used in this study are as follows: 0 mM (0 mM of root applied TU), 0.5 mM (0.5 mM of root applied TU) and 0.75 mM (0.75 mM of root applied TU).

Table 1 Summary of ANOVA regarding the effect of root-applied thiourea on growth, yield attributes, photosynthetic pigments, osmoprotectants and antioxidant enzyme activities of two potato cultivars under drought stress conditions.

Source of variation	df		Total Number of leaves/plant	Leaf area	RWC	Tubers fresh weight g/plant	Tuber dry weight g/plant	Chla	Chlb	
Cultivars	1		4.34*	1.82*	7.76***	3.31***	0.68**	0.78**	2.31***	
Drought Stress	1		69.44***	0.12***	0.02**	7.33**	0.86***	0.96***	6.33***	
Treatments	2		172.52***	0.33***	6.21**	16.92***	16.09***	26.09***	15.92***	
C × S	1		1.78 ns	0.01**	2.63***	0.42**	0. 24 ns	0. 24 ns	0.32**	
C × T	2		9.08 ns	5.01 ns	0.37 ns	0.49***	0.11 ns	0.11 ns	0.39***	
S × T	2		9.52 ns	0.01***	0.04 ns	3. 20**	0. 24 ns	0. 24**	2. 20**	
C × S × T	2		0.52 ns	5.18 ns	0.23 ns	0.03 ns	0. 29 ns	0. 29 ns	0.03 ns	
Error	24		4.47	0.001	0.13	0.03	0.08	0.08	0.03	
Source of variation	df	Carotenoids	Total
Chlorophyll	MDA	H2O2	RMP	Phenolics	Proline	Total soluble proteins	
Cultivars	1	0.04***	0.01*	126.29***	0.01 ns	60.83*	138.71**	5.26*	15.69**	
Drought Stress	1	3.67***	9.05***	76.73ns	1.24***	3,379.44***	27.56 ns	131.21***	138.96***	
Treatments	2	0.36***	34.83***	1,221.99***	4.99***	1,836.32**	88,880.02***	172.44***	162.05***	
C × S	1	0.01**	0.44*	45.37**	0.05**	20.55 ns	903.33***	2.45 ns	0.01 ns	
C × T	2	0.03***	0.28 ns	8.56 ns	0.00 ns	6.97 ns	122.02***	0.69 ns	1.48 ns	
S × T	2	0.01***	2.79***	15.19**	0.36***	15.28 ns	12.14 ns	3.92*	0.38 ns	
C × S × T	2	0.00**	0.70**	0.47 ns	0.01 ns	6.75 ns	211.85***	1.71 ns	3.94 ns	
Error	24	4.97	0.08	3.85	0.01	9.55	12.48	0.88	1.38	
Source of variation	df		CAT	POD	SOD	Anthocyanin	Total Soluble sugars			
Cultivars	1		0.94**	14.75**	84.99***	0.02 ns	4.77 ns			
Drought Stress	1		11,418.76**	26.17**	162.64***	5.59***	21,667.51***			
Treatments	2		18,437.13***	1262.34***	787.52***	5.47***	52,962.97***			
C × S	1		31.94 ns	56.34**	1.19 ns	0.14*	24.02 ns			
C × T	2		99.94*	25.91*	8.13 ns	0.15**	8.80 ns			
S × T	2		497.07***	32.00*	8.40 ns	0.05 ns	1,090.82*			
C × S × T	2		153.36*	23.16*	24.67*	0.01 ns	291.60			
Error	24		28.07	6.53	4.43	0.02	2.53			
Notes:

* 0.05 level.

** 0.01 level.

*** 0.001 level.

ns, non-significant; TNL, total number of leaves per plant; LA, Leaf area; TFW, Tubers fresh weight; TDW, tuber dry weight; SDW, RWC, Relative water content; Chl.a, chlorophyll a, Chl. b, chlorophyll b; Car., Carotenoids; Total. Chl., Total chlorophyll; MDA=, malondialdehyde; TSS, Total soluble sugars ; TSP, Total soluble proteins ; Pro, proline; H2O2, hydrogen peroxide; RMP, relative membrane permeability; TP, total phenolics; Anth, anthocyanins; SOD, superoxide dismutase; POD, peroxidase; CAT, catalase.

Changes in tuber fresh and dry weight per plant

The tuber fresh and dry biomasses were also measured in the present study in both cultivars of potato under the controlled (100% water field capacity) and drought stressed (65% water field capacity) environment with or without the foliar application of TU (Fig. 1; Table 1). According to the results the fresh biomass of tubers was decreased by 19% in potato-SH-5 and also decreased by 16% in potato-FD-73 when grown in the drought stressed environment compared to the control. However, the fresh weight was increased in both potato cultivars by applying the TU which significantly increased by 12% and 13% at the level of 0.75 mM respectively, compared to the plants which were not supplied by TU. Although, dry weight was also increased by 12.7% in potato-SH-5 and also decreased by 14.3% in potato-FD-73 at the level of 0.75 mM in the drought stressed environment, compared to those plants which were not supplied by the foliar application of TU.

Changes in relative water content and membrane stability characteristics

Drought stress significantly (P ≤ 0.01) decreased the relative water content (RWC) of both potato cultivars (Fig. 2; Table 1). However, the exogenous use of TU (0.5 and 0.75 mM) caused less of a decrease in the RWC. In this context, the RWC of cv. SH-5 was 0.58% and 0.25% and that of FD-73 was 3.97% and 3.20% at 0.5- and 0.75-mM TU, respectively, under drought stress conditions as compared with the control. In conclusion, TU at 0.75 mM was the most effective in decreasing the RWC. Cultivar differences were also apparent, as cv. SH-5 showed less of a decrease (1.42%) compared with cv. FD-73, which showed a significant decrease (3.39%). Membrane stability was reduced under drought stress in both cultivars: cv. SH-5 showed a decrease of 16.45%, and cv. FD-73 showed a decrease of 31.84%. Root application of TU significantly (P ≤ 0.01) reduced membrane stability in both cultivars compared to the control. In addition, the membrane stabilities of cv. SH-5 were 17.79% and 30.85%, and those of cv. FD-73 were 36.97% and 40.62% at 0.5- and 0.75-mM TU, respectively. Among the two potato cultivars, the cv. SH-5 showed less reduction in membrane stability compared with cv. FD-73, which showed a greater reduction (Fig. 2; Table 1).

Figure 2 Effect of root applied TU on (A) chlorophyll a, (B) chlorophyll b, (C) total chlorophyll, (D) carotenoids, (E) relative water content (F) and relative membrane permeability of leaves of two potato cultivars (Potato-SH-5 and Potato-FD.73-10) grown in controlled (100% water field capacity) and drought (65% water field capacity) stressed environment.

Means sharing similar letter(s) within a column for each parameter do not differ significantly at p < 0.05. Data in the figures are means of three repeats (n = 3) of just one harvest of potato plants ± standard deviation (SD). Different treatments (TU) used in this study are as follows: 0 mM (0 mM of root applied TU), 0.5 mM (0.5 mM of root applied TU) and 0.75 mM (0.75 mM of root applied TU).

Changes in photosynthetic pigments

Our results revealed that chlorophyll a and b, carotenoids, and total chlorophyll contents were significantly (P ≤ 0.001) reduced under drought stress conditions in both potato cultivars. Root application of TU significantly (P ≤ 0.001) increased photosynthetic pigments under drought stress and control conditions (Fig. 2; Table 1). Moreover, TU (0.75 mM) was most effective in increasing chlorophyll pigments, such as chlorophyll a by 25.98% and 19.12%, chlorophyll b by 5.82% and 35.29%, and total chlorophyll by 12.14 and 35.22% in cv. SH-5 and FD-73, respectively. There was also a significant (P ≤ 0.01) interaction between drought stress and TU treatment in chlorophyll a and b. Cultivar differences were also observed, such as in cv. SH-5 and FD-73, accumulation of chlorophyll a and b pigments increased by 25.13% and 12.15%, and by 28.89% and 38.21%, respectively. In conclusion, cv. SH-5 accumulated more chlorophyll pigments than cv. FD-73.

Changes in lipid peroxidation and hydrogen peroxide production

A non-significant increase was found in the MDA content of potato cultivars under drought stress conditions. Root application of TU significantly (P ≤ 0.001) decreased MDA accumulation under drought stress and control conditions (Fig. 3; Table 1). However, cultivars behaved differently as cv. SH-5 accumulated more MDA than cv. FD-73. The interaction between stress and TU treatment was also significant (P ≤ 0.01). A significant (P ≤ 0.001) increase in H2O2 content was observed in both cultivars under stressed conditions. Exogenous use of TU (0.5 and 0.75 mM) significantly (P ≤ 0.001) inhibited H2O2 accumulation under drought stress and non-stress conditions. However, 0.75 mM TU proved to be effective in decreasing MDA and H2O2 contents (Fig. 3; Table 1).

Figure 3 Effect of root applied thiourea on (A) malondialdehyde (B), hydrogen peroxide (C) anthocyanins, (D) phenolics of leaves of two potato cultivars under drought stress.

Mean with same letters (s) do not differ significantly at p < 0.05. Error bars above the means indicate standard error (n = 3)

Changes in the activities of antioxidant enzymes

The activities of CAT and POD were significantly (P ≤ 0.01) increased under stress conditions in both potato cultivars. The cultivar response was also significant (P ≤ 0.01), as cv. SH-5 showed increased activities of CAT and POD compared to those of cv. FD-5 under drought stress and control conditions (Fig. 4; Table 1). Exogenous application of TU significantly (P ≤ 0.001) further increased the activities of CAT and POD antioxidants. In this context, in cv. SH-5-, 0.5- and 0.75-mM TU increased CAT by 38.29% and 25.19%, and POD by 18.59% and 10.44%, respectively, whereas in cv. FD-73-, 0.5- and 0.75-mM TU increased CAT by 29.11% and 27.21%, and POD by 15.11% and 6.49%, respectively. TU at a concentration of 0.75 mM was found to cause greater changes than that at 0.5 mM. Similar to CAT and POD, SOD activity also increased (P ≤ 0.001) under drought stress conditions. However, root application of TU further increased the activity of SOD in both potato cultivars under drought stress and control conditions (Fig. 4; Table 1). Between the two TU levels tested (0.5 mM and 0.75 mM), 0.75 mM caused the greatest increase in the activities of antioxidants (SOD, POD, and CAT) under control and stress conditions.

Figure 4 Effect of root applied thiourea on (A) anthocyanins (B) total soluble proteins (C) total soluble sugars (D) catalase, (E) peroxidase (F) superoxide dismutase of leaves of two potato cultivars under drought stress.

Mean with same letters (s) do not differ significantly at p < 0.05. Error bars above the means indicate standard error (n = 3).

Changes in activities of phenolics and anthocyanins

A non-significant increase in phenolic content was found in both cultivars under drought stress conditions. Exogenous TU application significantly (P ≤ 0.001) increased the phenolic content in both potato cultivars (Fig. 3; Table 1). In this context, TU application increased the phenolic content of potato leaves in the cv. SH-5 by 6.98% and 2.97% and in FD-73 by 6.11% and 1.44% at 0.5- and 0.75-mM TU levels, respectively. Between the two levels, 0.75 mM proved most effective in increasing the phenolic content in both cultivars under stressed and non-stressed conditions. Anthocyanin content significantly (P ≤ 0.001) increased in both cultivars under drought stress: cv. SH-5 accumulated 95.56% and cv. FD-73 accumulated 60.1%. Root application of TU (P ≤ 0.001) increased anthocyanins in both the cultivars (43.28% and 30.28% in SH-5, and 27.38% and 21.15% in FD-73 at 0.5 and 0.75 mM TU levels, respectively). Cultivar response was also apparent, as cv-SH-5 accumulated more anthocyanins than cv. FD-73 (Fig. 3; Table 1).

Osmolyte accumulation

The results showed that proline content significantly (P ≤ 0.001) increased under drought stress conditions (Fig. 4; Table 1). Root application of TU significantly (P ≤ 0.001) further increased the proline content in stressed and non-stressed conditions in both cultivars. Meanwhile, 0.75 mM root-applied TU resulted in a higher proline content than that of 0.5 mM TU. Between the two cultivars, cv. SH-5 accumulated more proline than cv. FD-73. Total soluble sugars and total proteins were significantly (P ≤ 0.001) increased in both potato cultivars under drought stress conditions. However, exogenous use of TU (0.5 and 0.75 mM) significantly (P ≤ 0.001) further increased the accumulation of total soluble sugars and total soluble proteins under drought stress and control conditions (Fig. 4; Table 1). TU application at 0.75 mM resulted in greater accumulation compared to that at 0.5 mM. Cultivar differences were also recorded, and cv. SH-5 accumulated more total soluble sugars and total soluble proteins than cv. FD-73.

Relationship

Pearson’s correlation was used to illustrate the relationship between various growth, photosynthetic, and physiological parameters in the two potato cultivars (Fig. 5). However, both cultivars showed the same trend under drought-stressed and controlled environments with the exogenous application of TU. In potato-SH-5, the MDA content was negatively correlated with proline content, total soluble sugar, relative membrane permeability, plant height, total chlorophyll content, RWC, anthocyanin content, catalase activity, and total dry weight. Similarly, in potato-FD-73, the MDA content was also negatively correlated with proline content, total soluble sugar, relative membrane permeability, plant height, total chlorophyll content, RWC, anthocyanin content, catalase activity, and total dry weight. This relationship showed a close connection between various growth and physiological attributes of potato cultivars.

Figure 5 Schematic presentation interpreting the application of TU in two potato cultivars (Potato-SH-5 and Potato-FD.73-10) grown in controlled (100% water field capacity) and drought (65% water field capacity) stressed environment.

The drought stressed environment inhibited the plant growth characteristics and higher oxidative damaged was accumulated in the potato cultivars. Although, the negative impact of drought stress in both cultivars of potato can overcome by root applied by the application of TU, which not only increased plant growth characteristics and photosynthetic pigments but also decreased oxidative damage by enhancing antioxidant capacity in both cultivars of potatoes.

Principal component analysis

A principal component analysis was also performed to study the relationship between various growth and physiological parameters in potato cultivars (Fig. 6). In both potato cultivars, Dim1 and Dim2 exhibited the greatest contribution in the database. Dim1 contributed 72.5%, and Dim2 contributed 23.4% in the entire database. In combination, Dim1 and Dim2 occupy more than 95% of the entire dataset. The results also showed that all variables in both cultivars of potato were distributed similarly in the database. Among all the variables, proline content, total soluble sugar, relative membrane permeability, plant height, total chlorophyll content, RWC, anthocyanin content, catalase activity, and total dry weight showed a negative correlation, while MDA content showed a positive relationship among different variables in the database.

Figure 6 Correlation between various growth and physiological attributes in potato-SH-5 (A) and potato-FD-73 (B) grown under the drought stressed and controlled environment with the exogenous application of thiourea.

Different abbreviations are the figure are as follow: MDA, malondialdehyde content; Pro, proline content; TSS, total soluble sugar content; RMP, relative membrane permeability; PH, plant height; TC, total chlorophyll content; RWC, relative water content; Anth, anthocyanin content; CAT, catalase activity; TDW, total dry weight.

Discussion

The current study assessed the influence of root-applied TU on tuber yield, osmolyte accumulation, and enzymatic and non-enzymatic antioxidants in potato cultivars under drought stress conditions. Drought stress (65% field capacity) was observed to significantly (P ≤ 0.001) reduce the total number of leaves per plant and leaf area and tuber biomass in both potato cultivars compared with the control (Fig. 1; Table 1). Drought-induced reductions might be due to photosynthesis, respiration, cell extension, and enzymatic activities (Abid et al., 2018) because drought-stressed plants had a diminished number of leaves, and the development of new leaves, stems and leaflets, and leaf area were reduced compared to those in the control plants. This might be attributed to the impact of water stress on the physiological cycles in plants, such as photosynthesis, leaf zone extension, nucleic acid metabolism, protein synthesis, and partitioning of assimilates all being diversely effected (Khan et al., 2020b). In addition to this water deficiency, closed stomata, which ultimately reduces the production of photosynthesis, unfortunately becomes the primary driver in yield loss of plants (Li et al., 2020) because decreased photosynthesis restricts the mechanism of cell development and cell enlargement, eventually reducing yield (Yaseen et al., 2020). It has been documented that drought stress activates chlorophyll degradation in the leaves and reduces photosynthetic products (Khan et al., 2020a; Li et al., 2020); eventually, no leaves or leaf areas are affected during short water supply as plant systems require many photosynthetic end products for their growth (Zhao, Tan & Qi, 2007). Our results also demonstrated that the RWC and membrane stability of the leaves decreased significantly (P ≤ 0.01) in both potato cultivars under drought stress (Fig. 2; Table 1). With the enhancement of water stress levels, a decreased RWC and cell membrane stability index were observed. Bolat et al. (2014) reported that the decrease in leaf RWC and membrane stability index might be due to increased electrolyte leakage with the rise of water stress in both cultivars. This stress caused a reduction in the RWC, and the membrane stability index was significantly different from that of the control and other stressed plants.

Root-applied TU improved growth attributes, such as the increased in the leaf area in both potato cultivars grown under drought stress and control conditions (Fig. 1; Table 1). In the two potato cultivars (potato-SH-5 and potato-FD), TU application at 0.5 and 0.75 mM showed an increase in the total number of leaves and leaf area by compared to the control. Root application of TU is effective in stimulating growth through its direct interference with key enzymatic activities responsible for the biosynthesis and metabolism of growth-promoting substances (Abdelkader, Hassanein & Ali, 2012). Garg et al. (2005) also reported that TU-treated plants have an increased in the leaf area, compared to the control treatment. Many researchers found that TU enhanced growth in chickpea seeds under stressful conditions (Waqas et al., 2019). Similarly, the growth-promoting response to TU has also been studied in maize plants under stress conditions (Kaya et al., 2013). The TU-mediated increase might be attributed to the increased rate of photosynthesis, which added an adequate amount of photo assimilate and increased tuber weight (Baqer, Al-Kaaby & Adul-Qadir, 2020). A similar study was shared by Germchi, Behroozi & Badri (2011) in which TU treatment resulted in a higher weight of tubers and numbers. Ping et al. (2002) indicated that TU application plays a vital role in inducing tolerance against drought stress in both wheat cultivars (Fig. 2; Table 1).

The present study indicates that chlorophyll a and b, total chlorophyll, and carotenoid contents decreased significantly (P ≤ 0.001) in both potato cultivars when they were subjected to water stress (Fig. 2; Table 1). In the present study, it was shown that total soluble proteins, proline, and soluble sugar content increased significantly (P ≤ 0.001) under water stress conditions (Fig. 3; Table 1). These osmolyte-enhanced concentrations in drought-stressed plants could be involved in the cleaning of ROS, membrane permeability, osmotic adjustment, and maintenance in the actions of different enzymes in response to drought tolerance in plants (Naz et al., 2021), resulting in osmotic alteration (Yaseen, Zafar-ul-Hye & Hussain, 2019). The drought-induced decrease in chlorophyll might be attributed to a decline in the reduction in the synthesis of chlorophyll pigment complexes encoded by a cab gene family, a deterioration in light-harvesting structures of chlorophyll a and chlorophyll b protein complexes that play a role in protecting the photosynthetic apparatus (Sharma et al., 2020), or damage to lipids, pigments, and proteins of chloroplasts (Akram, Shahbaz & Ashraf, 2007). Similarly, Sehgal et al. (2017) documented a significant reduction in chlorophyll pigments observed in mustard plants under drought stress.

Our results revealed that TU in rooting medium boosted the chlorophyll content and accumulation of osmolytes in both potato cultivars under drought stress. TU-mediated improvement in photosynthetic pigments might be attributed to its positive effects on rubisco activity, which enhances the photosynthetic rate (Garg, Burman & Kathju, 2006) and the respiratory capacity of roots (Waqas et al., 2019). In addition, the positive effects of the TU-derived combination of protein kinases have been related to increasing the key regulation of cell division at various stages of cell development (Jagetiya & Sharma, 2013). The TU-assisted increase helps plants to develop osmotic adjustment by accumulating compatible solutes, such as soluble sugars, soluble protein, free amino acids, and total proline, which play key roles in defense mechanisms against drought stress (Fig. 3). Similar studies have shown that exogenous application of TU plays a favorable role in plant growth as it causes an enhancement in osmolyte accumulation under stressed conditions (Waqas et al., 2019).

Our recent study revealed an increase in MDA and H2O2 and also the activities of antioxidant enzymes, such as SOD, POD, and CAT in potato cultivars under drought stress conditions. Furthermore, H2O2 increased when drought triggered an increase in oxidative stress in potato cultivars (Fig. 3; Table 1). Studies related to our results found that drought stress increased the accumulation of MDA and H2O2 as already observed in wheat (Raza et al., 2013), cucumber (Wang et al., 2012), and canola (Akram et al., 2018). The intensity of oxidative stress in plants can be decreased by increasing the level of antioxidants, both enzymatic and non-enzymatic, such as catalase, peroxidase, superoxide dismutase, ascorbate peroxidase, ascorbic acid, and carotenoids and reducing glutathione, polyphenol oxidase, and glutathione reductase (Mohamed et al., 2020; Saleem et al., 2020b, 2020c). Our present study indicates increased antioxidant activity under drought stress conditions (Fig. 4; Table 1). A similar study by Akram et al. (2018) in canola and another by Shafiq, Akram & Ashraf (2015) reported increased activity of antioxidants under drought stress conditions.

Root-applied TU significantly reduced MDA and H2O2 accumulation in drought-stressed plants. The TU-induced reduction in MDA and H2O2 contents might be due to its potential to improve antioxidant defense as well as its role as a scavenger of ROS (Baqer, Al-Kaaby & Adul-Qadir, 2020). Similarly, it was previously reported that TU scavenges lipid peroxidation during drought stress conditions. Exogenous use of TU maintained the enzymatic capabilities of POD, SOD, and CAT in comparison to the plants without TU application. Similarly, it has been documented that TU induced increases in CAT and POD enzymes in stressful environments (Kaya et al., 2013). Moreover, TU increases the concentration of antioxidants and reduces the activity of ROS under salt stress conditions in mustard plants (Abdelkader, Hassanein & Ali, 2012).

In the present study, we found a non-significant increase in phenolic content and a significant increase in anthocyanin content in both cultivars of potato under drought stress conditions. In this regard, these non-enzymatic antioxidants detoxify ROS under oxidative stress (Nazar et al., 2020). Moreover, they serve as antioxidant compounds that can reduce the turgor loss of the cell membrane and decrease the cell lesions associated with stress. A lack of APX activity stimulates the production of anthocyanins to replace APX function and prevent cell damage (Mohamed et al., 2020). Similarly, an increase was observed in the total phenolic content in red beets under drought-induced stress (Naz et al., 2021).

In a recent study, TU application as a rooting medium amplified the phenolic content of potato leaves in cv. SH-5 by 6.98% and 2.97% and in FD-73 by 6.11% and 1.44% at 0.5 and 0.75 mM, respectively. Root application of TU (P ≤ 0.001) increased anthocyanin in both SH-5 (43.28% and 30.28%) and in FD-73 (27.38% and 21.15%) at 0.5 mM and 0.75 mM TU levels, respectively (Fig. 4; Table 1). Of the two, 0.75 mM was the most effective in increasing the phenolic content in both cultivars under stressed and non-stressed conditions. It has been reported that the TU-mediated increase in non-enzymatic antioxidants might be involved in detoxification of oxidative stress (Ping et al., 2002). Plants might increase the activity of antioxidant enzymes to allow them to survive in a stressed environment, which imposes significant energy costs (Waqas et al., 2019). The schematic presentation of whole experiment and their results in two varieties of potatoes grown in controlled and drought stressed environment is presented in Fig. 7.

Figure 7 Loading plots of principal component analysis on different studied attributes of potato-SH-5 (A) and potato-FD-73 (B) grown under the drought stressed and controlled environment with the exogenous application of thiourea.

Different abbreviations are the figure are as follow: MDA, malondialdehyde content; Pro, proline content; TSS, total soluble sugar content; RMP, relative membrane permeability; PH, plant height; TC, total chlorophyll content; RWC, relative water content; Anth, anthocyanin content; CAT, catalase activity; TDW, total dry weight.

Conclusions

The development, yield, and physiochemical function of potato plants were mainly affected by drought stress. The decline in the biochemical, physiological, and agronomic characteristics of plants was enhanced when the level of water stress was increased. In other words, the application of TU is beneficial for plants to adjust their growth and efficiency, and ultimately facilitates growth. Plants containing TU maintained the highest production of dry matter, leaf region, weight, and number of tubers compared to those in plants without TU. Under TU treatment, the plants demonstrated a lower decline in membrane stability, leaf RWC, chlorophyll content, carotenoid content, photosynthetic activity, and plant tuber yield. These plants showed less ROS as well as MDA and H2O2 contents and enhanced enzymatic action of POD, SOD, and CAT, with enhanced creation of completely dissolved sugars, dissolved proteins, and proline. The current study provides new information about the different impacts of TU on biochemical, morphological, and physiological characteristics in light of water stress in the potato. Such data can also provide practical information for future studies on improving drought tolerance and yield manageability in potato cultivation.

Supplemental Information

Supplemental Information 1 Raw Data.

Click here for additional data file.

Additional Information and Declarations

Competing Interests

Author Contributions

Data Availability

The authors declare that they have no competing interests.

Muhammad Hamzah Saleem analyzed the data, prepared figures and/or tables, authored or reviewed drafts of the paper, and approved the final draft.

Xiukang Wang analyzed the data, authored or reviewed drafts of the paper, and approved the final draft.

Abida Parveen conceived and designed the experiments, performed the experiments, authored or reviewed drafts of the paper, and approved the final draft.

Shagufta Perveen performed the experiments, analyzed the data, prepared figures and/or tables, and approved the final draft.

Saqib Mehmood performed the experiments, analyzed the data, prepared figures and/or tables, and approved the final draft.

Sajid Fiaz conceived and designed the experiments, analyzed the data, prepared figures and/or tables, and approved the final draft.

Sajjad Ali conceived and designed the experiments, analyzed the data, prepared figures and/or tables, and approved the final draft.

Sajjad Hussain analyzed the data, prepared figures and/or tables, and approved the final draft.

Muhammad Adnan analyzed the data, authored or reviewed drafts of the paper, and approved the final draft.

Naeem Iqbal performed the experiments, prepared figures and/or tables, and approved the final draft.

Aishah Alatawi conceived and designed the experiments, performed the experiments, prepared figures and/or tables, authored or reviewed drafts of the paper, and approved the final draft.

Shafaqat Ali analyzed the data, prepared figures and/or tables, and approved the final draft.

The following information was supplied regarding data availability:

The raw data are available in the Supplemental File.

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
