# Peer review of "Alleviation of drought stress by root-applied thiourea is related to elevated photosynthetic pigments, osmoprotectants, antioxidant enzymes, and tubers yield and suppressed oxidative stress in potatoes cultivars"

_PeerJ, doi:10.7717/peerj.13121_

## Round 0.1 · original submission · Major Revisions

Authors should extensively revise the manuscripts following the comments and suggestions of all three reviewers.

Reviewer 3 has requested that you cite specific references. You may add them if you believe they are especially relevant. However, I do not expect you to include these citations, and if you do not include them, this will not influence my decision.

·

Basic reporting

The manuscript is well-written

Experimental design

The experimental design is fine and methods are well described. However, authors have to clarify few issues as indicated in the general comments

Validity of the findings

The data are technically sound and conclusion is well stated.

Additional comments

General comments:
The manuscript entitled “Alleviation of drought stress by root-applied thiourea is related to elevated photosynthetic pigments, osmoprotectants, antioxidant enzymes, and tubers yield and suppressed oxidative stress in potato “ described the effect of thiourea application on growth and yield of potato under drought stress by assessing morpho-physiological and biochemical traits. Moreover, authors have demonstrated that thiourea application enhances antioxidant levels thereby reduces H2O2 and MDA level during drought stress in two potato genotypes. The finding of this study is very interesting and important for the readers from related field. In my opinion, the MS should be accepted for publication after addressing few important issues.

Specific comments:

Materials and methods
The authors considered 65% field capacity is drought stress. Is it true? Usually less than 60% field capacity is considered as light drought stress. Authors have to give references to justify that the moisture level is low enough too classified as light or moderate drought stress.

Additionally, author’s mentioned that seedlings were treated with rooting media of thiourea application (0.5 & 0.75 mM) at the vegetative stage. The question is how much thiourea solution was applied?

Authors have to mention the temperature and duration of drying of the potato samples for getting dry weight.

Results
The authors logically represented the data however the authors have not showed any images of plants for example how potato plants looks after 15 and 30 days of drought stress and how thiourea treatment recovers severity of drought phenotype?
Figure 2, 3, and 4 should be improved. In few cases, letter and standard error bar is overlapping. Letters should be plotted at the upper side of the standard error. It would be nice if the authors produces color bar graph.

Discussion
Few mistakes were found in using the abbreviations. Please carefully use the abbreviated name of ROS detoxifying enzymes.

Further recommendation
It would be pretty interesting and worth reading for the related readers if authors could produce a flow/process diagram on thiourea -mediated drought tolerance mechanism in potato based on the findings of the present study.

Reviewer 2 ·

Basic reporting

Authors must recognize what is thiourea in terms of environmental sustainability, not a favorable item.

Use of term:
Line 106: germination is not correct, sprouting or budding

Figure legend should be more compact and abbreviation shall be placed at other part of r legend legibility stand alone. descriptive to be more self-explanatory.

Reference section should be proof-read carefully. No sufficient citation info.
Line 476, 481, 498, 622: NO volume and pages or doi
Line 525 and 574: Style does not match with other citation

Experimental design

Overall, the data assessment itself looks neatly done on statistical approach using the correlation matrices for PCA.
However, more assessment can be made such with ontology mapping, could make it clear to conclude what authors aimed at.

Line 103: Authors mention "potato cultivars" SH-5 and FD73-110. What are they? The local varieties or breeding lines? How they are different in traits and pedigree. varietal genetic information is valid to analyze the data and lead to conclusion. Why they were chosen? Could it be with more varieties?

Validity of the findings

Authors claimed that there is positive effect for the environmental stresses in suing the direct observation of the use of the thiourea. But in looking practicality of how is the thiourea, it would not help agricultural practice. Thiourea can be high toxic and could be environmental toxin. Cyanamide could be processed from thiourea within human body if it is in-taken.
Hardly to see practicality, and while authors indicated their ethical screening, it would not be ethical in conducting research with negative value of chemical substances.

Additional comments

Experiment itself is done resonably, but the use of thiourea is dead end, and would not be practical for supporting abiotic stress.

Reviewer 3 ·

Basic reporting

1. MS need throughout English language checking.
2. The introduction is very extensive.
3. The references are very large in numbers and they are not correctly written as journal guide lines.
4.The figures are good in quality and visibility

Experimental design

1. Methodology and execute the experiment process is deprived. Data unwell interpreted and data presented in the manuscript about alleviation of drought stress by root-applied thiourea in potato are not having any new message about precise investigation.
2. Authors should follow uniform trend for cv. potato-SH-5 and potato- FD.73-110.
3. The methodology of estimation of antioxidant enzyme activities, MDA content and chlorophyll is not clear in material methods section.
4. For estimating antioxidative enzyme, the author should provide clear information about that whether extraction is done once or separately for each enzyme.
5. The author has calculated the activities of SOD, CAT and POD in same units. It should be written as i.e. µmol.min-1.mg-1 protein for CAT and POD.
6. What is the reason for extracted leaf chlorophyll with 80 % methanol instead of DMSO or acetone method?

Validity of the findings

1. The discussion is unwilling described by authors. The authors cited insufficient papers describing the effect of TU on potato and other crops. Several paper already describing the effect of plant bioregulaters under abiotic stress by
Nathawat et al. (2007), Biologia Plantarum 55 (1): 93-97;
Ramaswamy et al. (2007) Photosynthetica, 45 (3):477-480.;
Srivastava et al. (2008), Environmental Experimental Botany 64:250-255;
D’Souza et al. (2009) Acta Agronomica Hungarica 57(1): 21-31.
Nathawat et al. (2016), Experimental agriculture, 52(5): 418-433;
Nathawat et al.,(2018), Proc. Natl. Acad. Sci., India, Sect. B Biol. Sci. 88: 875-885.
Nathawat et al. (2021), Legume Research, 44: 67-73 and other workers. The effectiveness of TU and related compounds on crop plants should be cited in the manuscript. The discussion part of MS is tedious and inappropriate.

Annotated reviews are not available for download in order to protect the identity of reviewers who chose to remain anonymous.

---

## Round 0.2 · accepted · Accept

Both reviewers are satisfied with the revision of the manuscripts. However, authors must carefully check and improve the quality of English of the accepted manuscript during proofreading.

Reviewer 3 ·

Basic reporting

No Comment

Experimental design

No Comment, very Good

Validity of the findings

No Comment

Additional comments

No comment